# Alterations in Tumor Aggression Following Androgen Receptor Signaling Restoration in Canine Prostate Cancer Cell Lines

**DOI:** 10.3390/ijms25168628

**Published:** 2024-08-07

**Authors:** Demitria M. Vasilatis, Neelu Batra, Christopher A. Lucchesi, Christine J. Abria, Eva-Maria Packeiser, Hugo Murua Escobar, Paramita M. Ghosh

**Affiliations:** 1Department of Urologic Surgery, School of Medicine, University of California Davis, Sacramento, CA 95718, USA; dvasilatis@formerstudents.ucdavis.edu (D.M.V.); calucchesi@ucdavis.edu (C.A.L.); 2Veterans Affairs (VA)—Northern California Healthcare System, Mather, CA 95655, USA; nbatra@ucdavis.edu (N.B.); christine.abria@va.gov (C.J.A.); 3Department of Biochemistry and Molecular Medicine, School of Medicine, University of California Davis, Sacramento, CA 95718, USA; 4Department of Small Animal Medicine and Surgery, University of Veterinary Medicine Hannover, 30559 Hannover, Germany; eva-maria.packeiser@tiho-hannover.de; 5Department of Medicine, Medical Clinic III, Hematology Oncology and Palliative Medicine, University Medical Center Rostock, 18057 Rostock, Germany; hugo.murua.escobar@med.uni-rostock.de

**Keywords:** androgen receptor, androgen indifferent prostate cancer, dog, prostate cancer

## Abstract

In prostate cancer (PCa), androgens upregulate tumorigenesis, whereas in benign tissue, the revival of androgen receptor (AR) signaling suppresses aggressive behaviors, suggesting therapeutic potential. Dogs, natural PCa models, often lack AR in PCa. We restored AR in dog PCa to investigate resultant characteristics. Three AR-null canine PCa lines (1508, Leo, 1258) were transfected with canine wild-type AR and treated with dihydrotestosterone (DHT). In 1508, AR restoration decreased clonogenicity (*p* = 0.03), viability (*p* = 0.004), migration (*p* = 0.03), invasion (*p* = 0.01), and increased expression of the tumor suppressor *NKX3.1*, an AR transcriptional target (*p* = 0.001). In Leo, AR decreased clonogenicity (*p* = 0.04) and the expression of another AR transcriptional target *FOLH1* (*p* < 0.001) and increased the expression of *NKX3.1* (*p* = 0.01). In 1258, AR increased migration (*p* = 0.006) and invasion (*p* = 0.03). Epithelial–mesenchymal transition (EMT) marker (*Vimentin*, *N-cadherin*, *SNAIL1*) expression increased with AR restoration in Leo and 1258 but not 1508; siRNA vimentin knockdown abrogated AR-induced 1258 migration only. Overall, 1508 showed AR-mediated tumor suppression; AR affected proliferation in Leo but not migration or invasion; and EMT and AR regulated migration and invasion in 1258 but not proliferation. This study highlights the heterogeneous nature of PCa in dogs and cell line-specific effects of AR abrogation on aggressive behaviors.

## 1. Introduction

Prostate cancer (PCa) is the most frequently diagnosed cancer in men and ranks second in cancer-related deaths worldwide [1]. Despite its prevalence, PCa has very few naturally occurring animal models that adequately recapitulate the breadth of the disease, which is typically required for the preclinical testing and approval of novel therapies [2]. Dogs (i.e., *Canis lupus familiaris*; “canines”) are one of the few species that spontaneously develops PCa, but much remains unknown about the signaling cascades involved in their disease initiation and progression [3,4]. Moreover, little is known about the hallmark androgen receptor (AR) signaling pathway in dogs, the principal pathway targeted by therapeutics in PCa of men, and it is important to explore when investigating the use of dogs as an animal model for novel therapeutics [5,6,7].

PCa initially progresses through the canonical genomic androgen receptor (AR) signaling cascade, where AR remains sequestered by heat shock proteins (HSPs) in the cytosol of cells until it meets its cognate androgen ligand (e.g., testosterone or dihydrotestosterone [DHT]) before dimerizing and moving to the nucleus to act as a transcription factor. Here, AR promotes the expression of target genes by binding to their localization sequences called androgen responsive elements (AREs) and drives proliferation and differentiation. The actions of ARs can be modulated by co-activators and co-repressors that serve to enhance or dampen target gene expression [8,9]. This cascade is initially suppressed in PCa with androgen-deprivation therapies (ADTs) or chemical castration, which blocks androgen production or androgen binding to ARs, and tumors that respond to this therapy are considered androgen-sensitive [10].

Upon prolonged treatment with ADTs, a subset of PCa becomes resistant to this treatment [castration-resistant prostate cancer (CRPC)]; however, AR signaling continues irrespective of non-response to ADTs by various means, including gain-of-function mutations in ARs, copy number variations of ARs, mutations in co-repressors or co-activators of ARs, and others [11,12]. Treatment for CRPC comprises the chemotherapeutic agent docetaxel and androgen-receptor signaling inhibitors (ARSIs) which include the androgen synthesis inhibitor abiraterone acetate and the AR inhibitors enzalutamide, apalutamide, and darolutamide [7]. Following prolonged treatment with ARSIs, a smaller subset of patients with CRPC go on to develop other driver mutations and no longer rely on AR signaling, a subcategory of highly lethal PCa variants termed androgen-indifferent PCa (AIPC) where effective treatments are lacking [13].

Since the discovery and mainstream use of ARSI therapies, researchers have investigated the possibility that AR targeting has led to more aggressive, untreatable disease in a subset of patients and that the targeting of other pathways and even the reconstitution of AR signaling may be worth exploring [14,15]. It has been shown that ADT given at the “wrong” time enhanced, not inhibited, benign prostate growth, a phenomenon called the “Coffey Paradox” [16]. The normal function of the AR in the benign prostate is to suppress growth and induce terminal differentiation in benign prostate epithelial cells. Investigations determined that in benign cells, ligand-dependent AR binding to the c-Myc enhancer inhibits c-Myc transcription needed for proliferation, whereas in PCa, AR binding activates c-Myc transcription, thereby stimulating proliferation [17]. Therefore, ADTs increase cell growth in a benign prostate, whereas in hormone-sensitive PCa, they inhibit growth.

Additionally, although AR is known to promote PCa growth, it is also known to have a biphasic effect on cell growth (stimulation at 10^−12^–10^−9^ M [“low DHT”] and suppression at 10^−8^ M [“high DHT”]) and has been shown to transcribe genes that decrease DNA replication, repair, and synthesis [18]. Because of this, researchers have explored whether the revival of AR signaling in AR-indifferent or AR-null human PCa cell lines would abrogate aggressive behavior which may have broader implications for gene therapies in men with refractory disease [15,19,20]. While the majority of currently available PCa models reflect the stimulatory effects of ARs, very few demonstrate the inhibitory effects. Thus, additional models are required to study this aspect of PCa.

Studies revealed that neutered male dogs had a significantly increased risk for PCa [21]. Because dogs often have AR-null PCa, which may be due to their surgical castration at a young age, they may serve as a suitable animal model for this potential therapy. The aims of this study were to evaluate if AR reconstitution in AR-null canine PCa cell lines would abrogate aggressive behaviors in order to show that canine PCa utilizes this pathway in a similar way and may serve as a novel animal model for potential gene therapy in PCa for AR signaling restoration. Because PCa is a highly heterogeneous disease in humans as well as dogs, we hypothesize that AR revival will have varying effects on canine PCa that are cell line dependent.

## 2. Results

### 2.1. Androgen Receptor Gene Is Highly Conserved between Canines and Humans

The AR protein structure consists of an N-terminal domain (NTD), a DNA-binding domain (DBD), a hinge region (HR), and a ligand binding domain (LBD) [11]. Gene alignment between canines and humans resulted in 89.7% gene homology overall (Figure 1). There was 100% sequence conservation of the NTD motif ^23^FQNLF^27^, which is required for binding the NTD to the activation function 2 (AF2) region of the LBD as well as co-activators and allowing for the dimerization of AR [22] (Appendix A). There was also 100% sequence conservation of the NTD motif ^437^WHTLF^450^, which stabilizes ligand binding to ARs. There was 100% sequence conservation of the HR and DBD, which interacts with androgen response elements (AREs) of target genes [9]. The DBD also contains the nuclear localization sequence (NLS; 629RKLKKL634), which had 100% sequence conservation between the species. There was conservation of a coactivator binding site in the C-terminal domain (“LxxLL” motifs) [23]. The NTD glutamine repeats (amino acid abbreviation Q; DNA codon “CAG” or “CAA”) were less homologous between the species and in slightly different NTD locations (Appendix A), which is also a finding between different ethnicities of humans [24]. Therefore, the AR is considered similar in humans and canines.

### 2.2. Androgen Receptor Transfection and Treatment with DHT Results in Nuclear Localization and Expression of Downstream Target Genes in Canines

To examine the similarity of AR signaling in dogs compared to humans, we transfected three AR-null canine PCa cell lines (1508, Leo and 1258) with a pcDNA3.1(+) plasmid containing wild-type canine AR (pcDNA3.1-AR_can_) synthesized as described in Materials and Methods or with pcDNA3.1(+). Dogs, similar to humans, demonstrate a mean testosterone level of 2–5 ng/mL [25] and a mean dihydrotestosterone level of about 0.5–1.5 ng/dL [26]. The pcDNA3.1-AR_can_ transfected groups were either left untreated (+AR) or treated with 1nM DHT (+AR+DHT), while control groups were mock transfected. All three canine PCa cell lines showed successful protein expression of AR (Figure 2A) when compared to cell line matched controls. The ratio of AR levels to the loading control (lamin) is shown in Appendix A and demonstrates variations between AR levels in the three transfected lines. Moreover, all three canine PCa cell lines showed the successful translocation of ARs to the nucleus with DHT treatment (Figure 2B–D, right column, bottom image) and not in plasmid-transfected only groups treated with a vehicle (Figure 2B–D, right columns, top image) (enlarged pictures provided in Appendix A). Thus, all three canine lines, similar to human lines, have intact AR nuclear translocation pathways.

Next, we investigated whether ligand binding affected AR transcriptional activity in these cells. Despite significant increase in ARs in all cells upon transfection of the canine AR plasmid, DHT treatment did not further affect AR levels significantly in any cell line (Figure 2E). However, AR levels do not describe its activity. To investigate the effects of AR activation, we examined the effects of ARs and DHT on the expression of two well-known AR target genes—Nkx3.1 and FOLH1.

The well-characterized AR target tumor suppressor gene NKX3.1, which is expressed in canine prostate, was upregulated 1.5-fold with AR only and 4-fold upon DHT treatment in cell line 1508 (*p* = 0.001); it increased 3.5-fold upon AR transfection in Leo, and even further (*p* = 0.02) upon DHT treatment in AR-transfected cells compared to cells transfected with AR only. In contrast, cell line 1258 showed a 90% decrease in expression of NKX3.1 upon AR transfection, but its expression did not change further upon DHT treatment (*p* > 0.05) (Figure 2F). Given that NKX3.1 is considered to be a tumor suppressor, this suggests a tumor-suppressive function of AR transcriptional activity in 1508 and Leo but not in 1258.

Dogs, unlike humans, do not express prostate specific antigen (PSA), but FOLH1 (folate hydrolase 1), the gene encoding the enzyme prostate specific membrane antigen (PSMA), an AR target that is upregulated in human PCa, was also upregulated with AR restoration in 1508 (5-fold) and Leo (3.5-fold) but not with 1258. Upon DHT treatment, FOLH1 levels were unaffected in 1508 cells (*p* > 0.05) and significantly decreased when treated with DHT in cell line Leo (*p* < 0.01) while it remained unchanged in 1258 (Figure 2G). These results suggest that AR restoration in cell line 1508 and cell line Leo reduces aggressiveness but not in cell line 1258.

### 2.3. AR Signaling Restoration Decreases Proliferation in Canine PCa Cell Line 1508 and Leo but Decreases Metabolic Activity in 1258

Clonogenic assays were performed to determine whether AR restoration affects cell growth in canine PCa lines as has been shown in human PCa cell lines [15,18]. Restoration of AR signaling attenuated growth in cell lines 1508 (Figure 3A; *p* = 0.03) and Leo (Figure 3B; *p* = 0.04) but did not have an effect on cell growth in cell line 1258 (Figure 3C). Notably, 1258 had visibly increased growth rate with DHT treatment, though not significantly (*p* > 0.05). The effect of AR restoration on NADPH-dependent cellular oxidoreductase enzyme mediated metabolism, an indicator of viability, was investigated with MTT assays for all cell lines. The results support our above observation of tumor suppression by AR restoration in cell line 1508, whereas in cell line 1258, AR restoration increased the degree of tumor aggressiveness.

Flow cytometric assays for apoptosis and cell death were performed on all experimental groups for all three cell lines to determine whether the decrease in proliferation was due to transfection reagents or DHT treatment. There was no apoptosis detected in any of the control or experimental groups for any cell line (Figure 3G–I, quadrant 3 [Q3]). Though some cell death occurred in all groups, there was no substantial difference between the control groups and the experimental groups (Figure 3G–I, quadrant 1 [Q1]; *p* > 0.05). These results indicate that the change in cell growth rate estimated by clonogenic and MTT assays reflected an alteration in proliferation but not in apoptosis.

### 2.4. AR Signaling Restoration Decreases Migration in Canine PCa Cell Line 1508 and Leo but Increases It in 1258

To investigate whether restored AR signaling in canine PCa cell lines attenuates migration as has been shown in human PCa cell lines, wound closure assays were performed [27]. In both 1508 (Figure 4A, left column) and Leo (Figure 4B, left column) cell lines, over a period of 20 h, wounds were completely healed. In 1508 cells, AR expression slightly decreased wound closure rates over the same time period, but this difference was not significant, taking into consideration data over three independent experiments (Figure 4A, center column; *p* > 0.05). However, AR signaling revival by stimulation with 1 nM DHT attenuates migration in cell line 1508 over a period of 20 h (Figure 4A, right column; *p* = 0.006), indicating a decrease in aggressive behavior in these cells by restoration of AR signaling. In contrast, Leo’s migration over 20 h (Figure 4B, center column) was not significantly affected either by AR transfection or by 1 nM DHT treatment in the AR-transfected cells (Figure 4B, right column, *p* > 0.05). Therefore, cell migration in these cells was independent of AR signaling, indicating AR-indifferent behavior.

Contrastingly, cell line 1258 was much more slower moving compared to 1508 and Leo, resulting in only about 50% closure over a period of 20 h (Figure 4C, left column). These cells showed increased migration with AR transfection (Figure 4C, center column; *p* = 0.03) over the same period and even further (over a period of 20 h) when AR-transfected cells were treated with 1 nM DHT, suggestive of an aggressive phenotype induced by the restoration of AR signaling (Figure 4C, right column; *p* = 0.006). These results indicate that whereas in 1508 cells, AR attenuated migratory behavior, in 1258, it promoted migration, and in Leo cells, migration was independent of AR signaling.

It may be noted that both in cell line 1508 and in cell line 1258, there was no significant difference between the migration rate in the presence or absence of DHT; note that in all these experiments, the cells were cultured in fetal bovine serum (FBS), which has some level of androgens. Therefore, the effects of DHT are only evident when the effect is substantial. Small effects of androgens will not be visible simply by the addition of DHT.

### 2.5. AR Signaling Restoration Decreases Invasion and Markers of EMT in Some Canine PCa Cell Lines but Increases in Others

Boyden chamber invasion assays were performed to investigate if AR restoration in canine PCa cell lines attenuates invasion as has been demonstrated in human PCa cell lines [28]. Invasion was decreased in transfected and treated groups for cell line 1508 (*p* = 0.01) when compared to the control group (Figure 5A center column, right column). While there was slight decrease in invasion in transfected and treated groups for cell line Leo, it was not significantly different than the control group (*p* > 0.05) (Figure 5B). Cell line 1258 had decreased invasion with the presence of the AR plasmid (*p* = 0.01) (Figure 5C, center column); however, with DHT treatment and the restoration of AR signaling, invasion increased compared to the control group (*p* = 0.03) (Figure 5C, right column).

All experimental groups were evaluated for changes in the expression of EMT markers (i.e., SNAIL1, Vimentin, and N-cadherin) and compared to their respective control group. Vimentin was significantly upregulated with AR transfection in 1258 (2-fold) but not in 1508 or Leo; with DHT treatment, it increased further in cell lines Leo (*p* = 0.03) and 1258 (*p* = 0.03) but not in cell line 1508 (Figure 5D). With AR transfection, N-cadherin expression was unchanged in 1508, suppressed 65% in Leo, and increased 1.5-fold in 1258; with DHT treatment, it remained unchanged in 1508 (*p* > 0.05), increased further in Leo (*p* = 0.01), and decreased further in 1258 (*p* = 0.01) (Figure 5E). Lastly, SNAIL1 was suppressed by AR treatment in both 1508 (40% decrease) and Leo (60% decrease), but it increased 5-fold in 1258; upon DHT treatment, it was significantly upregulated in cell line 1258 (*p* < 0.0001) but remained unchanged in cell lines Leo and 1508 (*p* > 0.05) in comparison to cells transfected with AR alone (Figure 5F). AR restoration suppressed migration and invasion in 1508 but did not affect EMT, whereas in Leo, AR restoration upregulated EMT markers but did not affect migration or invasion. In contrast, AR restoration affected both invasion and EMT in 1258.

### 2.6. Vimentin Regulates Migration of 1258 but Not 1508

Metastasis is a serious problem in PCa and migration is the most obvious indicator of metastasis. EMT is known to stimulate migration and Figure 5 shows that cell line 1258, which experienced enhanced migration rates when AR signaling was restored, showed an increase in vimentin with AR signaling restoration. Since AR transfection and DHT treatment yielded opposite effects on the migration of cell lines 1508 and 1258 but had no impact on cell line Leo, we explored whether differences in vimentin or N-cadherin expression could explain the contrasting migration patterns in cell lines 1508 and 1258 by downregulating these genes using siRNA (Figure 6A,B). As before, AR signaling suppressed migration, as indicated by the wound closure assay (i.e., scratch assay), in cell line 1508 (Figure 6C) but increased it in cell line 1258 (Figure 6D). This pattern did not change in cell line 1508 when vimentin or N-cadherin was knocked down by siRNA, whereas in cell line 1258, the increase in motility by AR restoration was suppressed by the knockdown of vimentin and N-cadherin. Taken together, these results suggest that the upregulation of EMT proteins vimentin and N-cadherin, facilitated by AR restoration, mediates the increased migration potential of cell line 1258 through AR signaling.

A summary of the results described above is shown in Table 1.

## 3. Discussion

In this study, we explored the restoration of AR signaling in canine PCa cell lines to determine the similarity of this pathway between canines and humans as well as the potential use of canines as animal models for novel therapies in human PCa. We successfully transfected ARs into AR-null canine PCa lines and recreated AR signaling in all three cell lines in this study. It may be noted that we used transient transfections in all the studies—because of a lack of viability of PCa lines stably transfected with ARs unless the cells are cultured in androgen-free media. We found that multiple aggressive behaviors (i.e., proliferation, migration, and invasion) of cell line 1508 were abrogated by the revival of AR signaling, while in cell line Leo, these effects were significantly muted. In contrast, cell line 1258 became more aggressive with the restoration of AR signaling, with increased migration, increased invasion, and an increase in the expression of multiple EMT markers. These findings are important as they show the heterogeneity of physiology and response to AR restoration in canine PCa cell lines, which recapitulates the variabilities found in the physiology of different variants of human PCa and supports their use as a model for this dynamic disease.

The restoration of AR signaling was accomplished through transfection with a wild-type canine AR plasmid and treatment with a physiologic dose of DHT. Immunofluorescent staining indicated that ARs transfected into the cells without ligands retained a cytoplasmic localization but promptly translocated to the nucleus when DHT was added. These results indicate that despite a lack of AR expression and/or many of the AR target genes in PCa cells from castrated dogs, the AR signaling pathway was intact.

As the majority of male dogs in the Western world are castrated [29], dog PCa is most often low or null for the expression of ARs and AR target genes like *NKX3.1* [30]. Accordingly, all three canine PCa cell lines were initially AR-null, but AR was successfully expressed in the experimental groups (+AR or +AR+DHT) after transfection. The transfected ARs were shown to be sequestered to the cytoplasm without androgenic stimulation (+AR) but translocated to the nucleus with DHT treatment (+AR+DHT), a hallmark feature of all nuclear transcription factor signaling pathways when cognate ligands are present [9,31]. Due to a lack of canine-specific reporter gene assays, we were unable to conduct reporter gene assays to determine AR transcriptional activity on target genes—but we were able to determine the overall effect of ARs and DHT on a couple of AR transcriptional targets that indicate AR activity. The downstream target gene of AR, the tumor suppressor *NKX3.1*, was upregulated when AR signaling was restored in cell lines 1508 and Leo (*p* = 0.001; *p* = 0.01) but appeared nearly absent in cell line 1258, which is likely to be a specific feature of that cell line [32,33]. Studies show that staining for NKX3.1 protein is positive in the majority of primary prostatic adenocarcinomas, downregulated in many high-grade prostate cancers, and completely lost in the majority of metastatic prostate cancers [34]. *FOLH1* has been reported in dogs and is the gene that transcribes the protein PSMA in humans, also called FOLH1 in dogs; *FOLH1 is* a gene non-canonically repressed by ARs [35,36,37]. There was decreased expression of *FOLH1* with the restoration of AR signaling in cell line Leo (*p* = 0.0002) but no difference in expression in cell lines 1508 (*p* > 0.05) or 1258 (*p* > 0.05). Overall, cell lines 1508 and Leo displayed expected changes to AR targets with the restoration of AR signaling, but this was not visible in cell line 1258.

Abrogation of aggressive behaviors, including proliferation, has been shown to occur in human PCa cell lines with AR signaling restoration [15,18], and this was also explored in our study in three canine PCa cell lines. Although AR can promote PCa growth, it has also been demonstrated to be a potent tumor suppressor that inhibits proliferation by acting on genes that influence DNA replication, synthesis, modification, and repair (e.g., MCM7 [minichromosome maintenance complex gene], FANCI [Fanconi anemia complementation group gene]) by way of retinoblastoma protein (RB) recruitment, particularly when DHT is present [18]. For example, it is well known that the restoration of AR signaling in the human PCa-derived AR-null cell line PC3 attenuates cell growth rate, while the androgen-sensitive human PCa line LNCaP, that expresses a mutant AR, is growth-stimulated at low androgen levels but repressed at high levels of androgens [15,18]. To ensure that differences in cell number attenuation were not due to cell death or apoptosis, flow cytometry was performed for annexin V (apoptosis) and propidium iodide (cell death) for all cell lines and no difference was found between control and experimental groups (*p* > 0.05). Thus, the differences in cell numbers seen are likely due to changes in proliferation and not due to changes in cell death rates.

MTT assays were performed to evaluate whether AR restoration changes oxidoreductase metabolism (indicative of cell viability). Oxidoreductase metabolism decreased with AR signaling restoration in cell line 1508 but not in 1258. This was an interesting observation as AR-mediated PCa metabolism reduces glycolysis and enhances mitochondrial oxidative phosphorylation and lipogenic metabolism compared to non-cancerous prostate tissue [18,38,39]. Nonetheless, MTT assays more accurately represent cytosolic oxidoreduction by NADH and NADPH mechanisms and the restoration of AR signaling in cell line 1508 may have affected cytosolic metabolic activity [40,41]. It may be noted that in 1508, AR signaling attenuated cell numbers, whereas in Leo, the addition of DHT, but not AR expression, upregulated cell viability. Notably, Leo demonstrated the biphasic response seen in some human PCa, with an increase in cell numbers at 1 nM DHT but not at 10 nM DHT. In contrast, 1258 showed an increase in cell numbers with AR expression.

Others have reported that AR revival leads to decreased migration of human PCa by way of negatively regulating chemokines involved with migration, including several C-C motif ligand (CCL) and C-C motif receptors (CCR), like CCL2-CCR2 [27]. We found that only cell line 1508 had decreased migration upon AR signaling restoration. Cell line Leo did not have a significant change in migration with AR reconstitution while cell line 1258 had significantly increased migration when AR signaling was reconstituted. Furthermore, the transfected group (+AR) also had increased migration compared to controls. The increased aggressiveness, as demonstrated by increased migration, in cell line 1258 with restored AR signaling is a feature of CRPC where the presence of ARs leads to more aggressive disease [41].

AR signaling restoration has been shown to decrease the invasion of human PCa cell lines in numerous ways, including the reduction in adhesion to the extracellular matrix (ECM) and the modulation of genes involved in metastasis [15,42]. Our investigation of AR signaling restoration in canine PCa revealed variable results in the attenuation of invasion and was cell line-dependent. Invasion was abrogated in cell line 1508 with AR signaling restoration, although the expression of EMT markers (i.e., SNAIL1, Vimentin, and N-cadherin) was not significantly altered. Cell line Leo did not have a significant change in invasion with AR signaling restoration but interestingly had an increase in Vimentin and N-cadherin expression with AR signaling revival. Cell line 1258 not only showed increased invasion with restored AR signaling but also showed increased expression of Vimentin and SNAIL1 but not N-cadherin. Vimentin is a filamentous protein that provides structural and functional support to the cell and is overexpressed in aggressive PCa. The increase in N-cadherin expression is part of a dysregulated Wingless/Integrated (Wnt) signaling pathway, which has been shown to be further advanced by ARs in CRPC [43]. SNAIL1 is a downstream target of the activated PI3K/AKT/mTOR signaling pathway, which was shown to be overexpressed in multiple canine PCa cell lines [44].

Here, we show that the knockdown of Vimentin prevented AR-induced migration of cell line 1258 but had no effect on cell line 1508. Thus, Vimentin upregulation following AR restoration in cell line 1258 mediates the increase in AR-induced migration in these cells. Similar to human-derived PC3 cells, it is likely that the restoration of AR in cell line 1508 activated the “high DHT” end of the dichotomous response of AR signaling to androgens, which is repressive, rather than the “low DHT” end, which is stimulatory [15,18]. In that respect, cell line 1508 behaves most like a hormone-sensitive line when AR is restored. On the other hand, Leo may have pathway aberrations that reflect a cell line more similar to CRPC than androgen-dependent PCa when AR signaling is revived. Similar to Leo, the increase in aggressiveness with the restoration of AR signaling in cell line 1258 is most compatible with a cell line that resembles CRPC.

## 4. Conclusions

The results of the present studies confirm that, similar to previous publications utilizing human and rodent prostate epithelial cells [17], AR signaling is also suppressive of some canine prostate epithelial growth while stimulatory in others. PCa exhibits a complex response to AR signaling in dogs. The variability in responses and sensitivity to androgens is underscored by the contrasting migration patterns observed. Notably, cell line 1508 exhibits heightened sensitivity to hormones, resulting in decreased migration at physiological DHT levels (1 nM), whereas cell line 1258 displays reduced sensitivity and shows an increase in migration. This suggests that canine PCa, akin to human PCa, may demonstrate a biphasic response to androgen concentrations. It is interesting to note that while all four cellular functions studied here—clonogenicity, viability, migration, and metastasis—were abrogated by AR signaling in cell line 1508, indicating that a loss of AR expression was the major cause of tumorigenicity in these cells, in cell line Leo, AR affected only proliferative properties (clonogenicity and viability) but had no effect on migration or invasion, while in cell line 1258, ARs affected migration and invasion but did not affect proliferation. Thus, each cell line may serve as a model of a different aspect of PCa progression. Though canine PCa may serve as a model for human PCa, it is important to remember that it is a heterogeneous disease in canines as it is in humans, and each cell line may reflect different stages of PCa carcinogenesis.

## 5. Materials and Methods

### 5.1. Cell Lines and Cell Culture Materials

Canine prostate cancer cell lines 1508 and 1258 were generated by co-authors (E.M.P. and H.M.E.) from the University of Veterinary Medicine Hannover (TiHo), Hannover, Germany [45,46]; canine prostate cancer cell line Leo was purchased from Applied Biological Materials, Inc., Richmond, BC, Canada (Cat. No. T8278) [47]. All cell lines used were canine prostatic adenocarcinomas. All cells tested negative for Mycoplasma contamination. Cell lines were cultured in Roswell Park Memorial Institute (RPMI) cell culture medium 1640 (Invitrogen/Gibco, Carlsbad, CA, USA) and supplemented with 10% heat-inactivated fetal bovine serum (FBS) (Omega Scientific, Inc., Tarzana, CA, USA) and 100 U/mL penicillin-100 μg/mL streptomycin (Invitrogen/Gibco, Carlsbad, CA, USA). Treated cells had the aforementioned cell culture media supplemented with physiologic levels (1 nM) of dihydrotestosterone (DHT) (Cat. No. 521-18-6; Sigma Aldrich, St. Louis, MO, USA). Cells were kept at 37 °C in a humidified environment of 5% CO_2_ in air.

### 5.2. Androgen Receptor Plasmid Construction, Transfection, and Sequence Alignment

The DNA sequence encoding full-length canine androgen receptor from reference genome Dog10K_Boxer_Tasha was obtained from the Ensembl database (ENSCAFG00000016656.4; NCBI gene 403588) then synthesized and cloned between restriction sites BamH1 and Not1 in a pcDNA3.1(+)-C-HA vector (nom. nov. pcDNA3.1-AR_can_) (GenScript USA Inc., Piscataway, NJ, USA) [48]. Cell lines were transiently transfected for 8 h with the pcDNA3.1-AR_can_ plasmid using jetPrime^®^ DNA transfection reagent (Polyplus, Illkirch-Graffenstaden, France) according to the manufacturers’ instructions. Mock transfection was performed with an empty pcDNA3.1(+) vector (Cat. No. V79020; Thermo Fisher Scientific, Inc., West Sacramento, CA, USA). Groups transfected with pcDNA3.1-AR_can_ and then treated with 1 nM DHT for 24 h were considered the treatment group (+AR+DHT); groups transfected with the pcDNA3.1-AR_can_ plasmid without DHT treatment were considered the transfected group (+AR); and lastly, groups mock transfected with an empty vector were considered the control group (Ctrl). Protein sequence alignment was performed using an open access sequence alignment tool (EMBL-EBI, Hinxton, Cambridgeshire, UK, EMBOSS Water Pairwise Sequence Alignment) [49] pairing wild-type full-length 907 amino acid canine androgen receptor (ENSCAFG00000016656.4; NCBI gene 403588) to canonical wild-type full-length 920 amino acid human androgen receptor (ENSG00000169083; NCBI gene 9606).

### 5.3. siRNA Construction and Transfection

The canine siRNAs for Vimentin, N-cadherin, and Snail1 were constructed using Dharmacon siDESIGN center tool and purchased from Dharmacon (Dharmacon-Horizon Discovery, Cambridge, UK; custom siRNA design). All the siRNAs and the control siRNA were added at a final concentration of 25 nM for 48 h. Transfection was performed using Opti-MEM (Gibco ThermoFisher Scientific, 31985070) and Lipofectamine-2000 according to the manufacturer’s instructions. The sequence for each siRNA used in this study is provided in Table 2.

### 5.4. Cell Lysate and Protein Immunoblotting

Protein was extracted from cells grown for 3 days in RPMI supplemented media using 2X loading buffer (100 mM Tris-Cl pH 6.8; 4% (*w*/*v*) SDS; 0.2% (*w*/*v*) bromophenol blue; 20% (*v*/*v*) glycerol; 200 mM B-mercaptoethanol) [50]. Protein was quantitated by BCA assay (Pierce™ BCA Protein Assay Kit; Cat. No. 23225; Thermo Fisher Scientific, Inc.) and separated on 10% SDS-PAGE gels at 130 V for 1 h using minivertical electrophoresis cells (Mini-PROTEAN 3 Electrophoresis Cell, Bio-Rad, Hercules, CA, USA). Protein was transferred to 0.2 uM nitrocellulose membranes with the Trans-Blot Turbo transfer system (Bio-Rad, Hercules, CA, USA) for 30 min and then blocked with 5% nonfat dry milk in phosphate-buffered saline and 0.1% Tween 20 (PBST) for 1 h. Membranes were cut prior to incubation with primary antibody overnight at 4C. The following antibodies were used: AR (N-20; 1:1000; Santa Cruz Biotechnology, Santa Cruz, CA, USA) and lamin A/C (Cat. No. 2032; 1:1000; Cell Signaling Technology, Danvers, MA, USA). The next day, the membranes were washed with PBST three times for 10 min each and then incubated with secondary antibody conjugated to horseradish peroxidase (HRP) for 2 h. Development was performed using chemiluminescence (Pierce™ ECL Western Blotting Substrate; Cat. No. 32106; Thermo Fisher Scientific, Inc.) and membranes were imaged using a GE Amersham™ Imager 680 (GE Healthcare Life Sciences, Chicago, IL, USA). Gel loading was assessed by housekeeping protein lamin A/C (Cat. No. 2032; 1:1000; Cell Signaling Technology, Danvers, MA, USA).

### 5.5. Differential Gene Expression (RT-qPCR)

Total cellular RNA was prepared using the RNeasy kit (Cat. No. 47104; QIAGEN, Inc., Redwood City, CA, USA). cDNA was synthesized from 500 ng or 1 mg RNA using the iScript™ cDNA Synthesis Kit (Bio-Rad, Hercules, CA, USA) from three biological replicates. Real-time PCR was performed in triplicate using PowerUp™ SYBR™ Green Master Mix (Cat. No. A25741; Thermo Fisher Scientific) in triplicate. All aforementioned steps were performed according to the manufacturer’s instructions. HPRT1 was used as the endogenous expression standard. Data were collected on an Applied Biosystems 7500 Fast machine and analyzed using the relative standard curve method. The differential expression of various genes was compared between the control (Ctrl) and the transfected groups (+AR and +AR+DHT) of each cell line. Primers for each gene evaluated are provided in Table 3.

### 5.6. Immunofluorescence

Transfected cell lines were seeded at 10,000 cells per coverslip and were incubated for 24 h in media in a 37 °C CO_2_ incubator. Cells were then treated with vehicle or DHT for 24 h. After, cells were rinsed with PBST and then fixed to the coverslip with ice-cold methanol for 10 min on ice. Coverslips were then washed three times with PBST and then blocked with 10% goat serum for one hour at room temperature. Primary antibody (AR) was diluted 1:100 in 10% goat serum, applied to the coverslips and then incubated at 4C overnight in a humidity chamber. The next day, coverslips were washed three times with PBST and had anti-rabbit secondary antibody conjugated to rhodamine (1:500 in PBST; Life Technologies, Carlsbad, CA, USA) applied. Coverslips were then incubated in secondary antibody for 1 h at room temperature in the dark. After, coverslips were washed three times with cold PBST and coverslips were inverted and mounted onto uncharged glass slides with antifade mounting medium plus DAPI (Life Technologies, Carlsbad, CA, USA).

### 5.7. MTT Assay

Cells were grown in triplicate in 24-well plates at 50,000 cells per well and transfected and treated as abovementioned in Section 5.2. Following treatment, each well was incubated with 25 µL of 3-[4,5-Dimethylthiazol-2yl]-2,5-diphenyl-tetrazolium bromide (MTT; 5 mg/mL) (Sigma Aldrich, Burlington, MA) for 1 h in an incubator (37 °C, 5% CO_2_). The optical density (OD; 590 nm) was compared between control (Ctrl) and the transfected groups (+AR and +AR+DHT) of each cell line [54]. 

### 5.8. Clonogenic Assay

Clonogenic assays were prepared as previously described [55]. In short, cells were transfected and treated as abovementioned then plated in triplicate in a 6-well plate at 1000 cells per well. Media or media supplemented with DHT was refreshed every 48 h and all cells were allowed to grow for 14 days. Colonies were fixed and stained with 0.5% crystal violet. Total colony area (µm^2^) per well was measured to combat the tendency of some cell lines to make few large colonies versus others that make many smaller colonies. Colonies were measured and imaged with a BioTek Cytation 5 cell imaging multimode plate reader (Agilenty, Folsom, CA, USA) and the average area of 50 cells was calculated and set as a minimum threshold of detection. The total colony area was then compared between the control (Ctrl) and the transfected groups (+AR and +AR+DHT) of each cell line.

### 5.9. Flow Cytometry for Apoptosis

Cells were grown in 12-well plates at 100,000 cells per well in triplicate and transfected and treated as abovementioned in Section 5.2. Cells were conjugated to Annexin V and propidium iodide per the manufacturer’s instructions (FITC Annexin V/Dead Cell Apoptosis Kit; Cat. No. V13242; Thermo Fisher Scientific, Inc.). Flow cytometry was then performed on FACSAria (Becton Dickinson Immunocytometry Systems, San Jose, CA, USA) for cell lines 1508 and Leo and FACSCalibur (Becton Dickinson Immunocytometry Systems, San Jose, CA, USA) for cell line 1258. Cells were illuminated with 200 mW of 488 nm light or 635 nm light. Fluorescence was detected through a 630/22 nm (for PI) or 661/16 nm (for Annexin V-Alexa Fluor 647) band-pass filter. Frequency histograms were collected from 20,000 events and analyzed in FlowJo software version 10.8.1 (TreeStar, FlowJo LLC., Ashland, OR, USA).

### 5.10. Migration Assay

A migration assay was performed as previously described [56]. In brief, cells were grown in 24-well plates at 150,000 cells per well in triplicate and transfected and treated as abovementioned. Wells were at 100% confluency after the 24 h control or respective treatment. The monolayer was then linearly scratched with a p200 pipet tip. Wells were then washed with PBS and then media or media supplemented with DHT was added to the well. Time-lapse microscopy was used to acquire images every hour from the same field automatically over 24 h by a multimode plate reader (37 °C, 5% CO_2_; BioTek Cytation 5; Agilent). A masking algorithm was used to determine the wound confluency at 20 h relative to the original scratch wound’s diameter to combat variability in scratch wound diameters between replicates and cell lines.

### 5.11. Invasion Assay

The Boyden chamber invasion assay was performed as previously described [57]. Briefly, cells were grown in a 24-well plate in triplicate at 50,000 cells per well and transfected and treated as aforementioned. Then, cells were serum-starved for an additional 24 h. Transwell chambers (Cat. No. PTEP24H48; 0.8 µM pore size; Millicell^®^ 24-well hanging cell culture inserts; Millipore-Sigma, Burlington, MA, USA) were placed in a 24-well plate on ice, coated with 20 µL of Matrigel (2 mg/mL) with a 100 µL pipette tip, then incubated for 1 h (37 °C, 5% CO_2_). After, 100,000 cells from treatment and control groups were seeded in the upper chamber of the transwell in serum-starved media, while the lower chamber contained complete media. Cells were then allowed to migrate for 24 h. Inserts were decanted and the transwell was immersed in 4% paraformaldehyde (PFA) for 10 min at room temperature (RT). The excess PFA was then decanted and the transwells were then immersed in methanol for 10 min at RT. After, inserts were gently washed with PBS then immersed in 0.5% crystal violet for 30 min at RT. Inserts were then gently washed with water and the upper side of the transwell membrane was gently brushed with a cotton swab. Filters were allowed to dry overnight and then the underside of the filter was imaged. Five fields at low magnification were imaged and cells were then quantified and averaged single-blindedly by a veterinary pathologist (D.M.V.). The average number of cells that invaded were then compared between the control (Ctrl) and the transfected groups (+AR and +AR+DHT) of each cell line.

### 5.12. Statistical Analysis

Data were analyzed in GraphPad Prism (Boston, MA, USA) version 10.1.0. Normality was determined by the Shapiro–Wilk test. Differential gene expression was evaluated by Student’s *t*-test or ANOVA with Dunnett’s multiple comparisons test. Clonogenic formation area, MTT assay OD, migration assay end wound confluence, and invasion assay cellularity were compared between control and transfected groups with ANOVA with Dunnett’s multiple comparisons test. A *p*-value of <0.05 was considered significant.

## Figures and Tables

**Figure 1 ijms-25-08628-f001:**
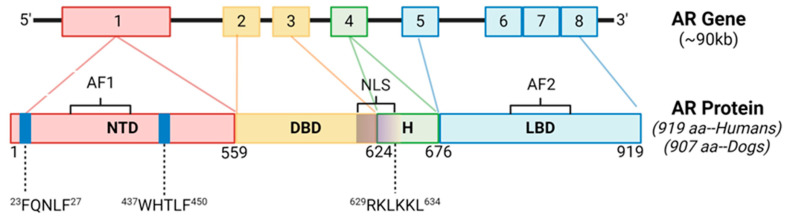
Androgen Receptor Gene Structure and Homology Between Canines and Humans. The androgen receptor (AR) is comprised of 8 exons. The N-terminal domain (NTD) contains sequences imperative for C-terminal binding during conformational changes in ARs after ligand binding, as well as ligand stabilization, and are 100% conserved between the species. The DNA binding domain (DBD) and nuclear localization sequence (NLS) are also 100% conserved between the species. AF1, activation function 1; AF2, activation function 2; H, hinge region; LBD, ligand binding domain.

**Figure 2 ijms-25-08628-f002:**
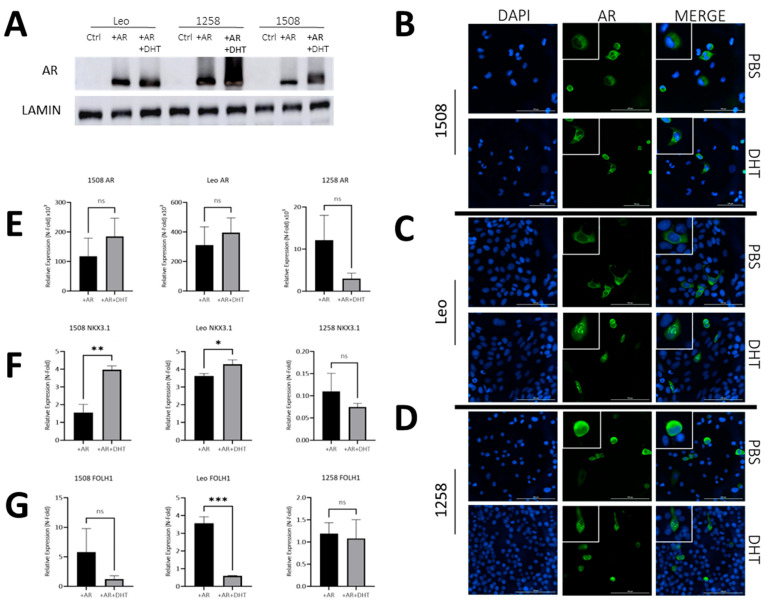
AR signaling restoration alters expression of downstream AR targets. (**A**) Protein immunoblot confirmed successful transfection and protein expression of canine AR. (**B**–**D**) AR transfection and DHT treatment leads to AR translocation to the nucleus, while without DHT treatment, ARs remain cytosolically sequestered. Bar= 100 µm. White boxes depict enlarged cells. (**E**,**F**) qPCR demonstrating effect of ARs and DHT on ARs and two of its transcriptional targets—Nkx3.1 and FOLH1. In each case, values for AR and AR+DHT treated cells were normalized to the corresponding value for control cells. Each bar represents mean ± S.D. for three independent readings. (**E**) qPCR shows successful AR gene overexpression after transfection in experimental groups relative to control groups. (**F**) NKX3.1 is a downstream target of ARs and was successfully upregulated via qPCR in cell lines 1508 (*p* = 0.001) and Leo (*p* = 0.01) but appears absent in cell line 1258. (**G**) FOLH1 is downregulated in the presence of ARs, which occurred in cell line Leo (*p* = 0.0002) via qPCR but not in cell lines 1508 or 1258 (*p* > 0.05). ns-not significant, * *p* < 0.05, ** *p* < 0.01, *** *p* < 0.001.

**Figure 3 ijms-25-08628-f003:**
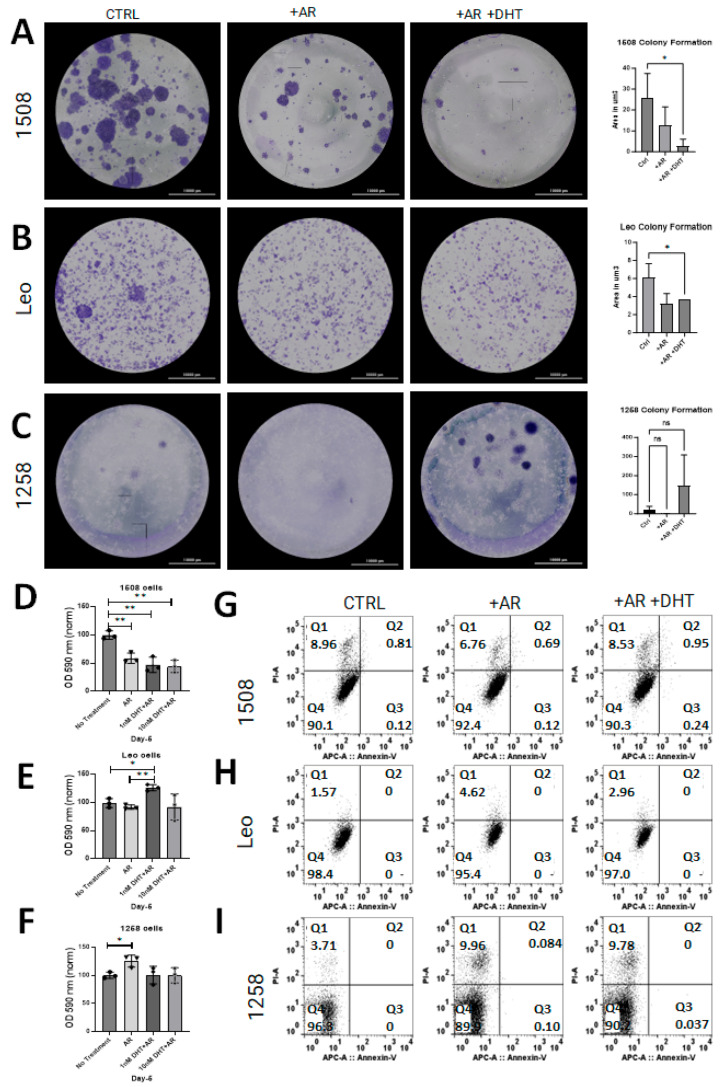
AR signaling restoration affects proliferation and metabolism. (**A**–**C**) Clonogenic assay to estimate rate of cell growth in cell lines 1508, Leo, and 1258 transfected with pcDNA3.1(+)-C-HA (CTRL) or with pcDNA3.1-AR_can_ (+AR) treated with vehicle (PBS) or 1 nM DHT (+AR+DHT). AR signaling revival attenuates proliferation in Leo (*p* = 0.04) and 1508 (*p* = 0.03) and appears to increase proliferation in 1258 but not significantly (*p* > 0.05). Scale bar-10,000 µm. (**D**–**F**) MTT assay for each cell line as indicated; oxidoreductase metabolism is unchanged with AR signaling revival in cell lines 1508 and Leo (*p* > 0.05) but decreased in cell line 1258 ((**F**); *p* = 0.04). ns–not significant, * *p* <0.05, ** *p* <0.01. (**G**–**I**) AnnexinV/PI flow cytometry for each cell line. Changes in proliferation assays was not secondary to cell death or apoptosis.

**Figure 4 ijms-25-08628-f004:**
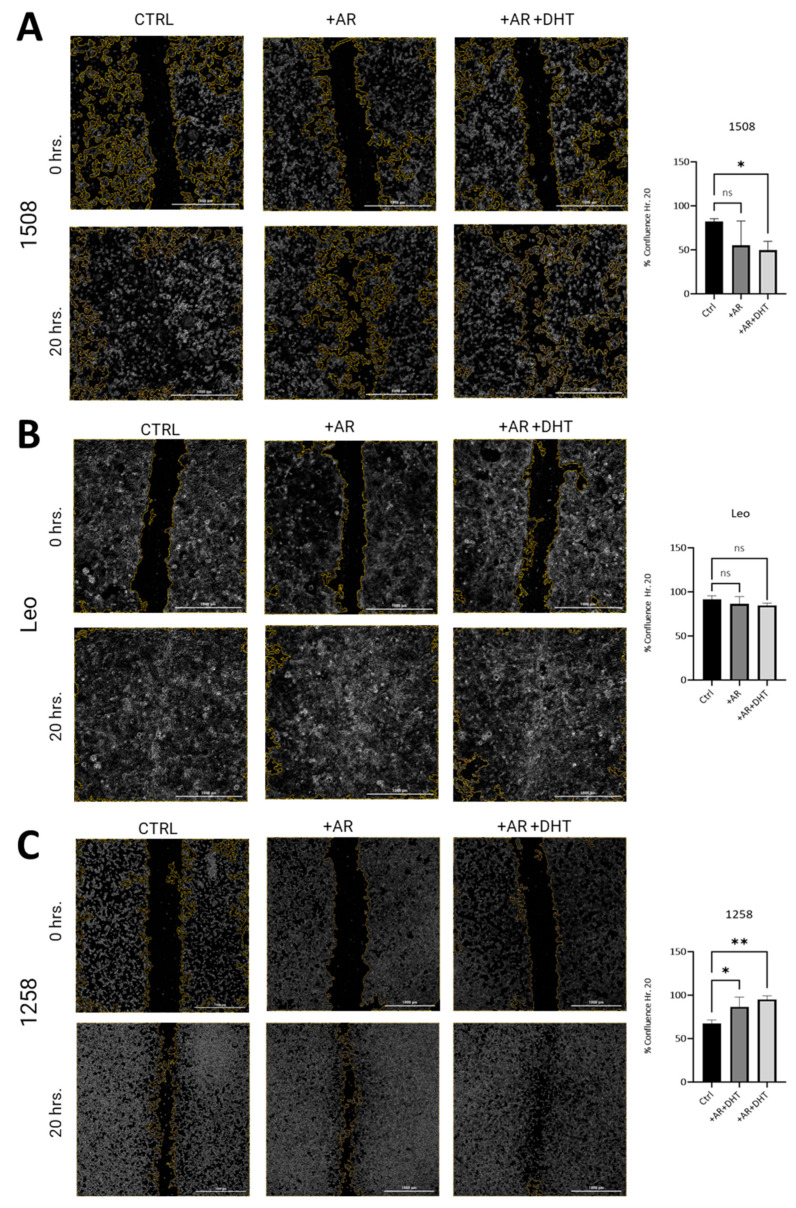
Scratch assay estimation of effect of AR signaling restoration on cell migration. (**A**) Cell lines 1508, (**B**) Leo, and (**C**) 1258 were transfected with an empty vector (Ctrl) or with wild-type canine AR, followed by treatment with vehicle (PBS) (+AR) or 1 nM DHT (+AR+DHT). Cells were wounded as described and allowed to grow back for 20 h. Rate of migration was estimated by % confluence after 20 h of culture. All experiments were conducted in triplicate, and data represent mean ± S.D. of three biological replicates. *p*-values represent comparisons as demonstrated in the accompanying graphs. ns – not significant, * *p* < 0.05, ** *p* < 0.01. Bar = 1000 µm.

**Figure 5 ijms-25-08628-f005:**
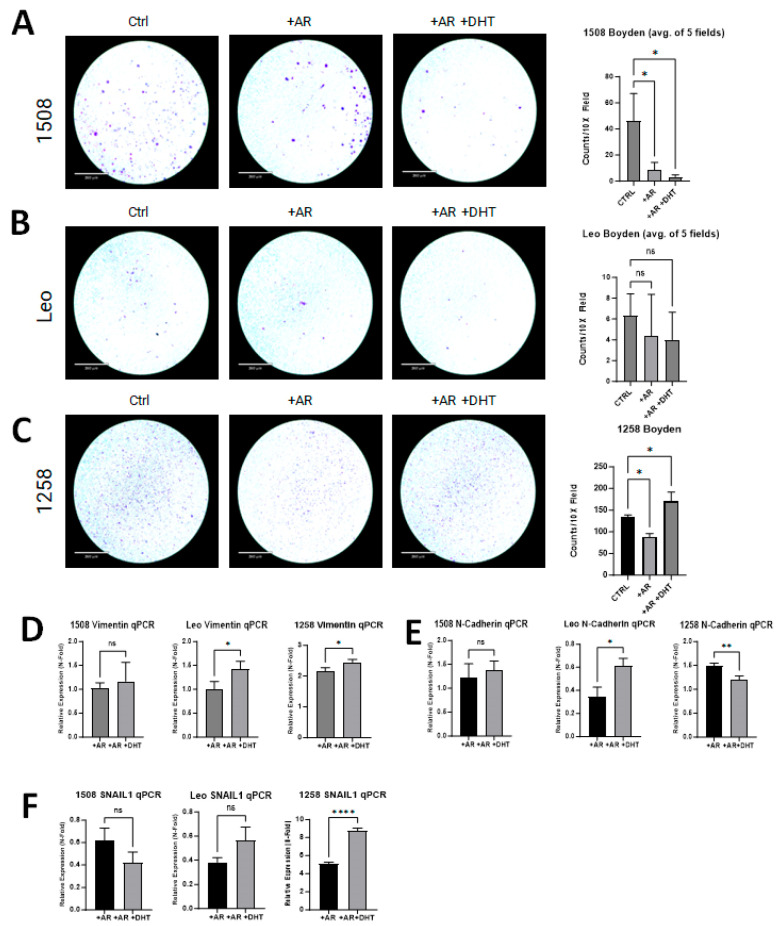
AR signaling restoration affects invasion and markers of EMT. (**A**) AR signaling revival attenuates invasion in 1508 (*p* = 0.01). (**B**) AR signaling restoration did not change the invasiveness of cell line Leo (*p* > 0.05). (**C**) Invasiveness decreased with restored AR expression in cell line 1258 but then increased further upon treatment with DHT. Scale bar = 360 µm. (**D**–**F**) qPCR results showing the expression of (**D**) Vimentin, (**E**) N-cadherin, and (**F**) SNAIL1 in 1508, Leo, and 1258 cells transfected with empty vector or with a plasmid expressing wild-type canine AR treated with PBS or 1 nM DHT. Data are represented as fold change over control (cells transfected with empty vector only). *p*-values represent comparison between DHT-treated and vehicle (PBS)-treated AR-transfected cells. All experiments were conducted in triplicate and data represent mean ± S.D. of three biological replicates. ns – not significant, * *p* < 0.05, ** *p* < 0.01, **** *p* <0.001.

**Figure 6 ijms-25-08628-f006:**
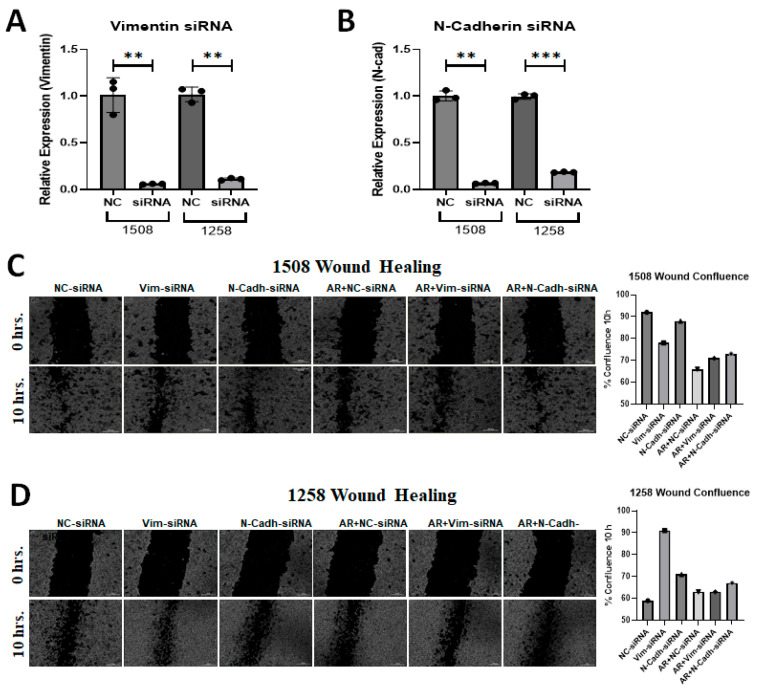
Silencing of EMT markers in canine PCa cells affects migration. (**A**) Expression of Vimentin following siRNA-mediated knockdown in (**left**) 1508 and (**right**) 1258 cells. (**B**) Expression of N-cadherin following siRNA-mediated knockdown in (**left**) 1508 and (**right**) 1258 cells. (**C**,**D**) Estimation of migration in (**C**) 1508 and (**D**) 1258 cells by scratch assay by Vimentin and N-Cadherin siRNA in cells transfected with an empty vector or with a plasmid expressing wild-type canine ARs. Scale bar = 300 µm. ns-not significant, ** *p* < 0.01, *** *p* < 0.001.

**Table 1 ijms-25-08628-t001:** Summary of findings in canine PCa cell lines after AR signaling restoration. Up arrow (↑) indicates increase, down arrow (↓) indicates decrease.

Aggressive Behavior	1508	Leo	1258
Colony Formation	↓	↓	No Change
MTT Metabolism	↓	↑	No Change
Migration	↓	No Change	↑
Invasion	↓	No Change	↑
*Vimentin*	No Change	↑	↑
*N-cadherin*	No Change	↑	↓
*SNAIL1*	No Change	No Change	↑

**Table 2 ijms-25-08628-t002:** The siRNA sequence used in this study for target gene knockdown.

Gene	siRNA Sequence (5′—3′)
*Vimentin*	Sense: GAAACUACAUGAUGAGGAAUU
	Antisense: UUCCUCAUCAUGUAGUUUCUU
*N-cadherin*	Sense: GAGAAGAAGACCAGGGAUUAUU
	Antisense: UAAUCCUGGUCUUCUUCUCUU
*SNAIL1*	Sense: GGACGAGGACAGUGGGAAAUU
	Antisense: UUUCCCACUGUCCUCGUCCUU

**Table 3 ijms-25-08628-t003:** Primers used in this study for target gene amplification.

Gene	Sequence (5′—3′)
*AR* ^a^	F: CGCCCCTGACCTGGTTT
	R: GGCTGTACATCCGGGACTTG
*NKX3.1* ^a^	F: TGAGGTGGTTGGAGGTTTGC
	R: TTTCATTGGCCCATCACTGA
*FOLH1* ^b^	F: GTGTTTGGTGGCATTGACC
	R: TTCTGCATCCCAGCTTGC
*Vimentin* ^c^	F: TACGCCAGCAATATGAAAGCG
	R: AGGGCATCATTGTTCCGGTTA
*N-cadherin* ^c^	F: AGCACCCTCCTCAGTCAACG
	R: TGTCAACATGGTCCCAGCA
*SNAIL1* ^d^	F: ACTGCAGCCGTGCCTTTG
	R: AAGGTTCGGGAACAGGTCTTG
*HPRT1* ^a^	F: AGCTTGCTGGTGAAAAGGAC
	R: TTATAGTCAAGGGCATATCC

Primers are from the following publications: ^a^, Rivera-Calderon et al. [51]; ^b^, Lai et al. [5]; ^c^, Yu et al. [52]; ^d^, Sammarco et al. [53].

## Data Availability

The data presented in this study are available on request from the corresponding author.

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
