# Peer review of "Alterations in Tumor Aggression Following Androgen Receptor Signaling Restoration in Canine Prostate Cancer Cell Lines"

_ijms, 2024, doi:10.3390/ijms25168628_

Round 1

Reviewer 1 Report

Comments and Suggestions for Authors

In the normal prostate, androgen receptor (AR) signaling is suppressive of the growth of the epithelial cells and in prostatic carcinogenesis there is a conversion of androgen receptor signaling from a growth suppressor in normal prostate epithelial cells to an oncogene in prostate cancer cells involves a gain of function in c-Myc regulation ( i.e. see   Isaacs JT Am J Clin Exp Urol 2018;6: 55-61; and Sena LA J Clin Invest 2022; 132(23)e162396). Thus, the results of the present studies have confined that like these previous publications utilizing human and rodent normal vs malignant prostate epithelial cells, AR signaling is also suppressive of normal canine prostate epithelial growth consistent with chacteristic loss of AR signaling by canine prostate cancers (CPca). What is different for the dog is that canine Pca doesn’t gain AR signaling as an oncogenic driver (i.e. see Bryan JN et al. Prostate 2007; 67:1174-81). The authors should include these previous references and incorporate them into the introduction as an appropriate starting point for their studies/results. Also since Myc expression has been established as one of the controlling elements in growth suppression vs. oncogenic stimulation of prostate epithelial cells, the authors should evaluate the level of Myc expression in their models plus and minus androgen.

Author Response

Reviewer 1

Comment 1: In the normal prostate, androgen receptor (AR) signaling is suppressive of the growth of the epithelial cells and in prostatic carcinogenesis there is a conversion of androgen receptor signaling from a growth suppressor in normal prostate epithelial cells to an oncogene in prostate cancer cells involves a gain of function in c-Myc regulation ( i.e. see Isaacs JT Am J Clin Exp Urol 2018;6: 55-61; and Sena LA J Clin Invest 2022; 132(23)e162396). Thus, the results of the present studies have confirmed that like these previous publications utilizing human and rodent normal vs malignant prostate epithelial cells, AR signaling is also suppressive of normal canine prostate epithelial growth consistent with characteristic loss of AR signaling by canine prostate cancers (cPCa). What is different for the dog is that canine PCa doesn’t gain AR signaling as an oncogenic driver (i.e. see Bryan JN et al. Prostate 2007; 67:1174-81). The authors should include these previous references and incorporate them into the introduction as an appropriate starting point for their studies/results.

Response 1: Thank you very much for these comments. These references do make this manuscript much stronger and bring a different perspective to the project. We have now incorporated the references you mentioned in the Introduction, as suggested (see references 16, 17, 21).

Comment 2: Also, since Myc expression has been established as one of the controlling elements in growth suppression vs. oncogenic stimulation of prostate epithelial cells, the authors should evaluate the level of Myc expression in their models plus and minus androgen.

Response 2: This is a fantastic idea, and we would love to do that – but these experiments would lead to additional studies that may be out of scope of the current manuscript. It is not just c-myc, but also other genes such as pRb, MCM7 and FANCI, as we note in the Discussion (and in our response to Reviewer 3 below), that may be involved in regulation of the biphasic effect. Hence (albeit reluctantly), we have decided not to include any c-Myc related studies (or pRb related studies, as suggested by Reviewer 3) in the current manuscript. However, we do intend to follow up with a thorough investigation of the role of c-Myc as well as pRb in canine PCa. Please look out for another manuscript on this topic in the near future.

Reviewer 2 Report

Comments and Suggestions for Authors

Dear authors of “Alterations in Tumor-Aggression Following Androgen Receptor Signaling Restoration in Canine Prostate Cancer Cell Lines”,

Thanks for your contribution to this field. This is an interesting article aimed at determining new cell models to study DHT effects in prostate cancer and its potential use as a therapy.

The manuscript is well written. Research is well organized and obtained results are convincing. The findings are of interest for prostate cancer in general.

Nevertheless, I notice some major points which need to be corrected before acceptance for publication.

1-      A cell model is step up on Canine cells and treated with a DHT level based on Human concentration. How is possible to compare DHT concentration values between Human and Canine? A point of discussion should be raised about this matter

2-      Two DHT concentration values have been evidenced: one named low 10^-12 10^-10 M which stimulate cell growth and one named high 10^-8 M which suppress cell growth. All the experiments are done at 10^-9 M which is just in between these two concentrations. It will be of interest if some of the major results will be addressed with these two concentrations: High and Low. This will increase the impact of the article and improve its discussion part.

Minor points:

1-      Figure 1, precisions for gene size are requested for Human and Canine to be homogenous with the protein part of the figure and with the text, lines 89-90

2-      Figure 2, the mock control is missing in all quantifications and its normalization to 1. It will help in result interpretations.

Looking forward seeing your modifications,

All the best,

Author Response

Comment 1: Thanks for your contribution to this field. This is an interesting article aimed at determining new cell models to study DHT effects in prostate cancer and its potential use as a therapy.

The manuscript is well written. Research is well organized and obtained results are convincing. The findings are of interest for prostate cancer in general.

Nevertheless, I notice some major points which need to be corrected before acceptance for publication.

  • A cell model is step up on canine cells and treated with a DHT level based on human concentration. How is possible to compare DHT concentration values between human and canine? A point of discussion should be raised about this matter

Response 1: Dogs, similar to humans, demonstrate a mean testosterone level of 2-5 ng/ml and a mean dihydrotestosterone level of about 0.5-1.5 ng/dl. We have added this information to the Results section (Section 2.2; lines 125-127).

  • Comment 2: Two DHT concentration values have been evidenced: one named low 10^-12 10^-10 M which stimulate cell growth and one named high 10^-8 M which suppress cell growth. All the experiments are done at 10^-9 M which is just in between these two concentrations. It will be of interest if some of the major results will be addressed with these two concentrations: High and Low. This will increase the impact of the article and improve its discussion part.

Response 2: Thank you for pointing this out. It was an error on our part to name low DHT as 10^-12 10^-10 M; it should actually be low 10^-12 10^-9 M. We have now corrected this definition (see lines 79-80). In Figure 3 D-F, we have compared two concentrations – 1 nM (low) and 10 nM (high). Of the three lines compared, Leo cells demonstrate the biphasic response the best – cell viability increases with 1 nM DHT but not with 10 nM DHT. We have now included this distinction in the text – both in Results (Section 2.3, lines 190-201) and in Discussion (lines 368-370).

Minor points:

  • Comment 3: Figure 1, precisions for gene size are requested for Human and Canine to be homogenous with the protein part of the figure and with the text, lines 89-90

Response 3: Thank you for these comments. This information is provided in Supplemental Figure 1.

  • Comment 4: Figure 2, the mock control is missing in all quantifications and its normalization to 1. It will help in result interpretations.

Response 4: Thank you for this comment. We fully appreciate your concern. However, all data was normalized to the mock control, so that would be at a numeric value of 1 in all the graphs. Given the scale of the y axis in most graphs, this information would not be useful – in fact, the resultant graphs would just increase the size of the figures without providing any additional useful data. However, we appreciate the fact that the lack of a control bar could be confusing, and as such have added text to the Figure Legend corresponding to these Figures explaining these calculations (see legend to Figures 2B-D, lines 147-150).

Reviewer 3 Report

Comments and Suggestions for Authors

This interesting article investigates AR signaling in PCa cells, revealing diverse responses across three canine PCa cell lines. While the results are comprehensive, the authors could discuss certain aspects more cautiously and suggest future research to strengthen their interpretations.

The use of transient transfection for AR overexpression and siRNA knockdown raises concerns about uniformity and potential for skewed results. Future studies might consider stable cell lines or gene knockout approaches, and quantifying transfection efficiency would provide valuable insights into AR's role in canine PCa. 

The results in Figure 2A (page 3) demonstrate significant differences in protein expression following AR overexpression across the three cell types, weakening the assertion of uniform signaling restoration. Future studies could quantify AR levels and assess transcriptional activity using reporter assays.

Overexpressing AR might lead to non-specific effects due to localization in unintended cellular locations and actions resulting from excessive protein levels. Dose-dependent experiments and comparisons with endogenous AR activation could help differentiate specific from non-specific effects. Techniques like ChIP-seq could also identify aberrant AR binding.

The study mentions biphasic effects of AR (page 2, lines 69-71), citing reference [16], which suggests that higher AR concentrations might lead to increased non-specific recruitment of Rb protein. However, the authors do not experimentally verify the involvement of Rb or discuss this aspect. Addressing this potential mechanism could provide valuable insights into the observed effects of AR overexpression.

The authors state that cell lines 1508 and 1258 were generated by co-authors, but no references are provided detailing their establishment. Could the authors cite relevant publications or provide information on the ethical approvals for obtaining the original canine tissues? This would enhance transparency and confirm adherence to animal welfare guidelines in the foundational work for this study.

Author Response

Comment 1: This interesting article investigates AR signaling in PCa cells, revealing diverse responses across three canine PCa cell lines. While the results are comprehensive, the authors could discuss certain aspects more cautiously and suggest future research to strengthen their interpretations.

The use of transient transfection for AR overexpression and siRNA knockdown raises concerns about uniformity and potential for skewed results. Future studies might consider stable cell lines or gene knockout approaches, and quantifying transfection efficiency would provide valuable insights into AR's role in canine PCa.

Response 1: Thank you very much for this comment. We are fully aware of the drawbacks of transient transfection of AR, but a stable AR transfection is not viable in any PCa cell line – either human or canine. We have previously stably expressed AR in human PC-3 cells, but those cells could only be cultured without loss of viability in androgen-free media. Addition of DHT to those cells instantly made them senescent. Nevertheless, we have tried to develop stable transfections in the canine lines with similar results. To prevent this, and to ensure the use of DHT in our experiments, we have used transient transfections. We have found that consistent use of similar reagents for similar periods of time provided remarkably similar results, hence we were confident of the results obtained. In addition, stable cell lines are also prone to artefacts arising from prolonged gene regulations that affect downstream targets and adversely affect cell metabolism. Therefore, transient transfections are used to overcome those problems. As the reviewer suggests, we state in the discussion the problems with stable transfections (lines 302-305).

Comment 2: The results in Figure 2A (page 3) demonstrate significant differences in protein expression following AR overexpression across the three cell types, weakening the assertion of uniform signaling restoration. Future studies could quantify AR levels and assess transcriptional activity using reporter assays.

Response 2: Yes, Figure 2A does show that each cell had a different level of AR transfection. As the reviewer has suggested, we have now quantified AR levels, and these results are included in Supplementary Figure 2. There is no AR transcriptional activity assay conducted, mainly because the reporter gene assay substrates for canine AREs currently do not exist and would take a much longer time to develop and test than reasonable. However, AR transcriptional activity was assessed by expression of AR target genes. Figure 2F shows mRNA expression of Nkx3.1, an AR target gene that is strongly upregulated by AR in hormone sensitive PCa, both canine and human. Both 1508 and Leo cells showed an upregulation of Nkx3.1 when AR transfected cells were stimulated by DHT; however, 1258 cells, which resemble CRPC, showed no significant change. On the other hand, FOLH1, a gene that encodes the protein PSMA, also expressed in dogs and humans, that is known to be AR repressed, was downregulated by DHT in AR-positive cells in 1508 and Leo but not in 1258. Based on these results, we conclude that AR transcriptional activity is active in 1508 and Leo cells but not in 1258. We apologize for not making this clear in the original version, but we have added these details in the text under Results (section 2.2, lines 138-143).

Comment 3: Overexpressing AR might lead to non-specific effects due to localization in unintended cellular locations and actions resulting from excessive protein levels. Dose-dependent experiments and comparisons with endogenous AR activation could help differentiate specific from non-specific effects. Techniques like ChIP-seq could also identify aberrant AR binding.

Response 3: This is a valid concern; however, Figure 2B, C and D demonstrate that the AR ONLY localizes to two places – nucleus and cytoplasm – in the canine PCa cells. Transfection of AR leads to expression of AR, but that AR is localized in the cytoplasm, as it is inactivated. Activation of AR with DHT causes AR translocation to the nucleus. This is illustrated in all three canine PCa cell lines. We have now emphasized these observations in the Discussion (lines 315-320; 324-327).

As we emphasize in our response to Reviewer 2, we have indeed conducted studies with two different DHT concentrations and show the differential effect of DHT on cell viability (Figure 3D-F).

We agree that ChIP-Seq would indeed identify aberrant AR binding in these cells. This is a fantastic idea that will indeed make the difference between the cells explicit. However, this would give rise to a number of other issues – regarding the identification of target genes, that are beyond the scope of the current study. As mentioned earlier, we are planning follow up publications, where these ideas will certainly be followed up on.

Comment 4: The study mentions biphasic effects of AR (page 2, lines 69-71), citing reference [16], which suggests that higher AR concentrations might lead to increased non-specific recruitment of Rb protein. However, the authors do not experimentally verify the involvement of Rb or discuss this aspect. Addressing this potential mechanism could provide valuable insights into the observed effects of AR overexpression.

Response 4: Thank you for this important comment. Reviewer 1 has also mentioned the possibility of c-Myc being a mediator of AR’s effects. As is obvious, there are many other mediators of cell functions regulated by the AR, not just pRb and c-myc (referred in that citation). As stated in our response to Reviewer 1, focusing on c-myc or pRb in the current study will detract from the current focus, which is to show the heterogeneity of response of various canine PCa lines to AR restoration. As mentioned above, the mechanisms will be elucidated in a follow up publication that we are already planning. The current paper will demonstrate only the reliance of these cells on EMT markers vimentin and N-cadherin. We do not want to dilute that message by introducing other factors such as pRb and c-Myc. These factors will be elaborated on in follow up papers that are already being planned.

We actually do mention pRb in the manuscript, please see Discussion (lines 348-350).

Comment 5: The authors state that cell lines 1508 and 1258 were generated by co-authors, but no references are provided detailing their establishment. Could the authors cite relevant publications or provide information on the ethical approvals for obtaining the original canine tissues? This would enhance transparency and confirm adherence to animal welfare guidelines in the foundational work for this study.

Response 5: Thank you for pointing out this oversight. Thanks to this comment, we have now included the relevant references in Materials and Methods (references 46, 47 and 48).

Round 2

Reviewer 1 Report

Comments and Suggestions for Authors

acceptable